# Potent and Broad-Spectrum Bactericidal Activity of a Nanotechnologically Manipulated Novel Pyrazole

**DOI:** 10.3390/biomedicines10040907

**Published:** 2022-04-15

**Authors:** Silvana Alfei, Debora Caviglia, Alessia Zorzoli, Danilo Marimpietri, Andrea Spallarossa, Matteo Lusardi, Guendalina Zuccari, Anna Maria Schito

**Affiliations:** 1Department of Pharmacy (DIFAR), University of Genoa, Viale Cembrano, 16148 Genoa, Italy; spallarossa@difar.unige.it (A.S.); matteo.lusardi@edu.unige.it (M.L.); zuccari@difar.unige.it (G.Z.); 2Department of Surgical Sciences and Integrated Diagnostics (DISC), University of Genoa, Viale Benedetto XV, 6, 16132 Genoa, Italy; debora.caviglia@edu.unige.it (D.C.); amschito@unige.it (A.M.S.); 3Cell Factory, IRCCS Istituto Giannina Gaslini, Via Gerolamo Gaslini 5, 16147 Genoa, Italy; alessiazorzoli@gaslini.org (A.Z.); danilomarimpietri@gaslini.org (D.M.)

**Keywords:** CR232-loaded dendrimer NPs, 3-(4-chlorophenyl)-5-(4-nitrophenylamino)-1H-pyrazole-4-carbonitrile (CR232), Gram-positive and Gram-negative MDR isolates, MICs and MBCs, time-kill experiments, cytotoxicity on human cells, selectivity index

## Abstract

The antimicrobial potency of the pyrazole nucleus is widely reported these days, and pyrazole derivatives represent excellent candidates for meeting the worldwide need for new antimicrobial compounds against multidrug-resistant (MDR) bacteria. Consequently, 3-(4-chlorophenyl)-5-(4-nitrophenylamino)-1H-pyrazole-4-carbonitrile (CR232), recently reported as a weak antiproliferative agent, was considered to this end. To overcome the CR232 water solubility issue and allow for the determination of reliable minimum inhibitory concentration values (MICs), we initially prepared water-soluble and clinically applicable CR232-loaded nanoparticles (CR232-G5K NPs), as previously reported. Here, CR232-G5K NPs have been tested on several clinically isolates of Gram-positive and Gram-negative species, including MDR strains. While for CR232 MICs ≥ 128 µg/mL (376.8 µM) were obtained, very low MICs (0.36–2.89 µM) were observed for CR232-G5K NPs against all of the considered isolates, including colistin-resistant isolates of MDR *Pseudomonas aeruginosa* and *Klebsiella pneumoniae* carbapenemases (KPCs)-producing *K. pneumoniae* (0.72 µM). Additionally, in time–kill experiments, CR232-G5K NPs displayed a rapid bactericidal activity with no significant regrowth after 24 h on all isolates tested, regardless of their difficult-to-treat resistance. Conjecturing a clinical use of CR232-G5K NPs, cytotoxicity experiments on human keratinocytes were performed, determining very favorable selectivity indices. Collectively, due to its physicochemical and biological properties, CR232-G5K NPs could represent a new potent weapon to treat infections sustained by broad spectrum MDR bacteria.

## 1. Introduction

Across the past two decades, the quantity of MDR bacteria has grown dramatically worldwide [1,2]. MDR pathogens are defined as bacteria that are resistant to at least three classes of antibiotics and represent one of the biggest threats to global health and food security [3,4]. Bacteria’s resistance to antibiotics occurs when pathogens, by different mechanisms, change their response to such drugs [5]. MDR bacteria belong to both Gram-positive, Gram-negative, and Mycobacteria groups [6]. They are responsible for a wide range of infections that are becoming progressively more difficult to treat, as the antibiotics used to cure them become less and less, or even no longer, effective [7].

Antibiotic resistant bacteria can affect anyone, of any age, in any country in the world, causing an increasing number of deaths in hospitals, long-term care facilities, and community settings [8]. Additionally, to counteract infections sustained by MDR bacteria, longer hospital stays, and higher medical costs are required, which, consequently, has a negative impact on the economic burden of individuals, families, and societies [9]. New resistance mechanisms are emerging and spreading globally, pressuring our ability to treat common infectious diseases [10,11]. In this context, it has been recently reported that the dissemination of antimicrobial resistance through horizontal gene transfer could be facilitated by commonly used nanoparticles (NPs), such as Ag, CuO, and ZnO NPs, as well as their ion forms (Ag^+^, Cu^2+^, and Zn^2+^) [12]. Paradoxically, NPs widely adopted for years as antibacterial agents could be responsible for the increase of the natural transformation rate in bacteria by stimulating the stress response and ATP synthesis [12].

Worryingly, without effective and urgent action, we are heading towards a post-antibiotic era, where common infections and minor injuries can once again kill [13,14]. In this regard, the lack of effective antibiotics against the most dangerous species of MDR bacteria seriously puts the achievements of modern medicine at risk. In fact, practices such as organ transplants and surgeries certainly become much more dangerous without antibiotics capable of preventing and treating infections.

We must also consider that, although antibiotic resistance can occur naturally, the misuse of antibiotics in humans and animals accelerates the process and, therefore, behaviour changes that include actions to reduce the spread of infections through vaccination, hand washing, practising safer sex, and good food hygiene are needed [15]. 

Preventing and controlling the spread of antibiotic resistance is a concern that should affect society at all levels [15]. Individuals, policy makers, health professionals, the healthcare industry, and the agricultural sector should work and cooperate closely to limit the diffusion of MDR bacteria [8]. Particularly, while the health industry should invest in research and development of new antibiotics, vaccines, and new diagnostic tools, the agricultural sector should administer antibiotics to animals only under veterinary supervision and should not use antibiotics for growth promotion or to prevent diseases in healthy animals [16].

Tackling antibiotic resistance is a high priority for the World Health Organization (WHO), which has led to multiple initiatives that address antimicrobial resistance, including the World Antimicrobial Awareness Week, the Global Antimicrobial Resistance Surveillance System (GLASS), and the Interagency Coordination Group on Antimicrobial Resistance (IACG) [8]. Furthermore, the Global Antibiotic Research and Development Partnership (GARDP) is a joint initiative of WHO and the Drugs for Neglected Diseases initiative (DNDI), which encourages research and development through public–private partnerships [8]. 

However, although monitoring antibiotic resistance is essential [6,15], the search for new antibacterial agents, which act by means of mechanisms different from those of available antibiotics and that have a lower propensity to develop resistance, represents one of the greatest challenges for researchers [6]. 

In particular, the Antimicrobial Availability Task Force of the Infectious Diseases Society of America, FDA, and other organizations have highlighted the urgent need to develop new antibiotics with activity against Gram-negative organisms, including *Pseudomonas aeruginosa* [17]. Meanwhile, colistin (polymyxin E) is often the only treatment option available, and it is increasingly used as the last line of therapy against Gram-negative “superbugs”, despite its renal and neurologic toxicity [17]. Unfortunately, the increasing emergence of strains that are also resistant to colistin, including *P. aeruginosa, Acinetobacter baumannii*, and *Klebsiella pneumoniae*, makes the situation dramatic.

In this alarming scenario, consisting of poor therapeutic options whose activity will probably not endure, the five membered, metabolically stable, heterocyclic diazole ring of pyrazole, having a demonstrated effectiveness on different bacterial strains including MDR variants by targeting different metabolic pathways of both Gram-positive and Gram-negative bacteria [18], could represent an excellent template molecule to develop new antibacterial agents that are effective where traditional antibiotics fail. 

We recently developed a novel one-pot, low-cost synthetic strategy for the preparation of highly functionalized pyrazole derivatives [19]. As such, the pharmaceutical relevance of pyrazoles, the in-house availability of a reliable synthetic protocol for their preparation, and the global need of new therapeutic options against infections by MDR bacteria prompted us to study the antibacterial properties of CR232 (Figure 1). This compound has been selected based mainly on the structural similarity with BBB4, a 3-phenyl pyrazole recently reported as having antibacterial properties, which were further improved by its formulation in water-soluble dendrimer NPs [6]. 

Preliminary microbiologic investigations were carried out to determine the MICs of CR232; however, due to its total water-insolubility and its tendency to precipitate in the aqueous medium of the experiments, the results obtained were not completely reliable and the resulting MICs were difficult to interpret, with only MICs ≥ 128 µg/mL being assumed. Therefore, to perform further biological evaluations and possibly hypothesize a future clinical application of CR232, we solved the solubility issues of CR232 by two nanotechnological approaches, using both a dendrimer and liposomes as encapsulating and solubilizing agents [20]. In both cases, CR232-loaded NPs with enhanced water-solubility and properties suitable for in vivo administration were achieved [20]. 

Here, the NPs obtained by using G5K, a lysine-containing dendrimer, as a solubilizing agent (CR232-G5K NPs) were selected to be evaluated for their effects on both bacterial and normal human cells. A preliminary screening showed that CR232-G5K NPs possessed a remarkable antibacterial activity against MDR bacteria that were representative of both Gram-positive and Gram-negative species. Therefore, we have studied the antibacterial and bactericidal effects of CR232-G5K NPs on several clinical isolates of different species of both families, obtaining excellent results. Interestingly, CR232-G5K NPs were also effective against isolates of *P. aeruginosa* and *K. pneumoniae* that are resistant to colistin, for which all other clinically approved antibiotics, and even recently developed cationic macromolecules acting as the cationic antimicrobial peptides, failed [21]. 

Next, once the strong and broad-spectrum antibacterial and bactericidal activity of CR232-G5K NPs were established in order to evaluate the feasibility of their clinical application, their cytotoxicity on human keratinocytes was evaluated. In parallel, G5K and CR232 were also tested under the same conditions for comparative purposes.

### Why CR232-G5K NPs and Not a Liposome-Based Formulation?

CR232-G5K NPs were chosen due to their higher water-solubility, drug loading capacity (DL%), encapsulation efficiency (EE%), dendrimer structure, and cationic character, which are features reported to support antibacterial effects [22]. In particular, the cationic nature of CR232-G5K NPs would promote their interaction with the negative bacterial surface, thus favouring the localization and accumulation of NPs on prokaryotic cells. Additionally, the high DL% of CR232-G5K NPs, would allow for the release of large amounts of the transported pyrazole at the target bacteria upon administration of a low micromolar dose of the formulation. Moreover, it is now generally accepted that cationic macromolecules, including dendrimers, thanks to electrostatic interactions with the bacterial surface, cause the depolarization of the membrane and its progressive permeabilization through the formation of increasingly large pores [22]. In our case, pore formation would have favoured the entry of CR232 into the bacterial cell. 

## 2. Materials and Methods

### 2.1. Chemical Substances and Instruments

The synthetic procedure for preparing CR232 and the polyester-based cationic NPs loaded with CR232 (CR232-G5K NPs) used in this study were recently reported [19,20]. In addition, experimental details and characterization data concerning CR232-G5K NPs are available in the Appendix A). Further, the dose-dependent cytotoxicity experiments performed with G5K on eukaryotic ovarian cancer cells (HeLa) using a PAMAM-NH_2_ dendrimer as a control and the related results are available in Appendix A.

### 2.2. Microbiology

#### 2.2.1. Bacterial Species Considered in This Study

Several clinical isolates of Gram-positive and Gram-negative species, for a total of 36 strains, were employed in this study. All bacteria belonged to a compendium obtained from the School of Medicine and Pharmacy of the University of Genoa (Italy). Their identification was carried out by VITEK^®^ 2 (Biomerieux, Firenze, Italy) or the matrix-assisted laser desorption/ionization time-of-flight (MALDI-TOF) mass spectrometric technique (Biomerieux, Firenze, Italy). In particular, 21 strains were of Gram-negative species, including four isolates of MDR *P. aeruginosa*. Among these, one strain was resistant to colistin, one to the combination avibactam-ceftazidime, and one was an MDR strain isolated from a patient with cystic fibrosis. The other Gram-negative strains were comprised of three isolates of *Escherichia coli*, of which one produced β-lactamase enzymes of the KPC family and one produced New Delhi metallo carbapanemases (NDMs), four isolates of MDR *Stenotrophomonas maltophilia*, four strains of *Klebsiella pneumoniae* carbapenemases (KPCs)-producing *K. pneumoniae* (one of which was also resistant to colistin), and three isolates of MDR *Acinetobacter baumannii*. Finally, one *Morganella morganii*, one *Providencia stuartii*, one *Proteus mirabilis*, and one *Salmonella* group B were considered among the Gram-negative bacteria of this study. Fourteen strains were Gram-positive, including six isolates of the genus *Enterococcus* (3 vancomycin resistant (VRE) *E. faecalis* and three *E. faecium*, of which one vancomycin susceptible (VSE) and two VRE), one sporogenic *B. subtilis*, and seven clinical isolates of the genus *Staphylococcus*. In particular, four isolates were methicillin-resistant *S. aureus* (MRSA) and three were methicillin-resistant *S. epidermidis* (MRSE), one of which was also resistant to linezolid.

#### 2.2.2. Determination of the Minimal Inhibitory Concentrations (MICs)

The antimicrobial activity of CR232-G5K NPs was investigated, determining their MICs following the microdilution procedures detailed by the European Committee on Antimicrobial Susceptibility Testing EUCAST [23], as also reported in our previous studies [17,21]. Here, serial two-fold dilutions of a solution of CR232-G5K NPs (DMSO), ranging from 1 to 128 μg/mL, were used. All MICs were obtained in triplicate, the degree of concordance in all the experiments was 3/3, and the standard deviation (±SD) was zero. 

#### 2.2.3. Time-Kill Experiments

Killing curve assays for CR232-G5K NPs were performed on various isolates of *S. aureus, E. coli*, and *P. aeruginosa* as previously reported [17,24]. Experiments were performed over 24 h at concentrations four times that of the MICs.

### 2.3. Evaluation of Cytotoxicity of CR232, G5K, and CR232-G5K NPs on Human Keratinocytes 

#### 2.3.1. Experimental Protocol for Cell Culture

Human skin keratinocyte cells (HaCaT), derived from a generous gift of the Laboratory of Experimental Therapies in Oncology, IRCCS Istituto Giannina Gaslini (Genoa, Italy), were grown as a monolayer in RPMI 1640 medium supplemented with 10% fetal bovine serum (*v*/*v*), 1% penicillin-streptomycin, and 1% glutamine (Euroclone S.p.A., Milan, Italy), cultured in T-25 cm^2^ plastic flasks (Corning, NY, USA) and maintained at 37 °C in a 5% CO_2_ humidified atmosphere. Cells were tested and characterized at the time of experimentation as previously described [25].

#### 2.3.2. Viability Assay

HaCaT cells were seeded in 96-well plates (at 4 × 10^3^ cells/well) in complete medium and cultured for 24 h. The seeding medium was removed and replaced with fresh complete medium that had been supplemented with increasing concentrations of empty dendrimer (G5K), CR232, or CR232-G5K (0 μM, 1 μM, 5 μM, 10 μM, 15 μM, 20 μM, 25 μM, 50 μM, 75 μM, or 100 μM). Cells (quadruplicate samples for each condition) were then incubated for an additional 4, 12, or 24 h. The effect on cell growth was evaluated by the fluorescence-based proliferation and cytotoxicity assay CyQUANT^®^ Direct Cell Proliferation Assay (Thermo Fisher Scientific, Life Technologies, MB, Italy) according to the manufacturer’s instructions. Briefly, at the selected times, an equal volume of detection reagent was added to the cells in culture and incubated for 60 min at 37 °C. The fluorescence of the samples was measured using the mono-chromator-based M200 plate reader (Tecan, Männedorf, Switzerland) set at 480/535 nm. The experiments were carried out at least three times and samples were run in quadruplicate.

### 2.4. Statistical Analyses

The statistical significance of differences between the experimental and control groups in cytotoxicity studies was determined via a two-way analysis of variance (ANOVA) with the Bonferroni correction. The analyses were performed with Prism 5 software (GraphPad, La Jolla, CA, USA). Asterisks indicate the following *p*-value ranges: * = *p* < 0.05, ** = *p* < 0.01, and *** = *p* < 0.001. The results have been reported in Appendix A). Concerning MIC values, experiments were made in triplicate and the concordance degree was 3/3 and ±SD was zero.

## 3. Results and Discussion

### 3.1. Brief Recapitulation of the Main Characteristics of CR232-G5K NPs

Table 1 collects the main characteristics of CR232-G5K NPs that were determined, reported, and discussed in our previous work [20].

### 3.2. Antibacterial Effects of CR232-G5K NPs

Most of the pharmacological activities of the pyrazole nucleus have been studied since the year 1944, but the first studies on the antimicrobial effects of pyrazole derivatives begun only after the year 2000 [6]. 

Today, their antimicrobial properties are extensively documented [6,18]. Among the developed molecules containing the pyrazole ring, some have been demonstrated to possess broad-spectrum antibacterial activity and significant antibacterial effects against MDR isolates of *A. baumannii*, MRSA, and VRE. As reported, the antimicrobial properties of the molecules containing the pyrazole ring would be due to the presence of amino groups in the structure [26]. Assuming that the pyrazole derivative recently synthetized by our group (CR232) could also be a good candidate as new antibacterial agent, after having formulated it as dendrimer NPs for reasons explained in the introduction to this study, the obtained CR232-loaded water-soluble NPs have been tested here against several isolates of different species of Gram-positive and Gram-negative bacteria.

#### 3.2.1. Determination of MIC Values

The values of MICs and of minimum bactericidal concentrations (MBCs) were determined on several clinical isolates of different species of Gram-negative and Gram-positive bacteria for a total of 36 isolates, and the results obtained are given in Table 2 (Gram-positive) and Table 3 (Gram-negative). 

According to the results reported in Table 2 and Table 3, since the MIC value of 128 µg/mL was assumed as the cut-off value above which to consider the compound inactive, CR232-G5K NPs showed wide antibacterial profiles and were found to be inactive (MICs > 128 µg/mL) against only three out of the 36 strains, i.e., *P. mirabilis, M. morganii*, and *P. stuartii*, which represent particularly difficult-to-treat isolates of Gram-negative species. CR232-G5K NPs appeared to be visibly active against three MRSA, two out of four KPCs-producing strains of *K. pneumoniae*, and one out of four MDR *P. aeruginosa* that are also resistant to the combination avibactam-ceftazidime (MICs = 2.89 µM), while it proved to be very potent against all other Gram-negative (MICs = 0.72–1.44 µM) and Gram-positive bacteria tested, displaying the best antibacterial effects on *Enterococci* (MICs = 0.36–1.44 µM)*, S. epidermidis* (MICs = 0.36 µM), and *B. subtilis* (MIC = 0.36 µM). In all cases, MBC values equal to, or superimposed on MIC values, were observed. The macromolecular complex containing CR232 displayed micromolar MIC values even lower than those observed for a potent cationic copolymer reported to have a remarkable broad-spectrum activity [21].

Furthermore, CR232-G5K NPs showed very low MIC values against a MDR strain of *P. aeruginosa* isolated from a cystic fibrosis patient (MIC = 0.72 µM), against both a MDR strain of *P. aeruginosa* and a strain of KPCs-producing *K. pneumoniae* that are also resistant to colistin (MICs = 0.72 µM), and against a strain of MDR *P. aeruginosa* that is also resistant to the avibactam-ceftazidime combination (MIC = 2.89 µM) clinically used to counteract bacterial resistance to carbapenems [27]. The activity of CR232-G5K NPs against these particularly drug-resistant strains of *P. aeruginosa* and *K. pneumoniae* is particularly relevant considering that these pathogens represent frightening superbugs responsible for severe nosocomial infections associated with dramatic outcomes [17].

To our knowledge, except for BBB4-G4K NPs recently developed and essayed by us as antibacterial agents for future clinical applications [6,28], there are only two other studies in the literature on the polymer formulation of pyrazole derivatives [26,29] and only one of those reports the development of pyrazole-based polymers for antibacterial purposes, although for textile finishing and not for therapeutic uses [26]. In this regard, the antibacterial activity of CR232-G5K NPs showed a significantly extended spectrum of action in comparison with that of BBB4-G4K NPs [6], resulting in it being active against most of both Gram-positive and Gram-negative isolates that were considered herein. Additionally, compared to BBB4-G4K NPs, CR232-G5K NPs showed good potency against MRSA, while against MRSE they were 10-fold more potent. Therefore, with this new study, we have obtained a significant improvement in the potential of our pyrazole-based dendrimers as potential antimicrobial agents.

As previously determined [20], CR232-G5K NPs contain CR232 31.7% (*w*/*w*) and after 24 h, they release 99.3% of the entrapped CR232 into the physiological medium. Based on these data, the MIC and MBC values of the free CR232 released in the medium by the amounts of NPs that inhibited the bacterial growth were estimated and reported in Table 2 and Table 3 (columns 4 and 5). These data were essential to compare the antibacterial effects of CR232 delivered by CR232-loaded NPs with those of pristine CR232 (MICs ≥ 128 µg/mL) and those of other small molecules containing the pyrazole nucleus previously assayed as antibacterial agents. As summarized in Table 2 and Table 3, the MIC values estimated for the released CR232 were in the range of 5.05–20.2 µg/mL on Gram-positive bacteria (except for three MRSA isolates (MICs = 40.4 µg/mL)), and in the range of 10.1–20.2 µg/mL on Gram-negative bacteria (except for *P. mirabilis, M. morganii*, and *P. stuartii* (MICs > 40.4 µg/mL), for two out of four KPCs-producing *K. pneumoniae*, and for one out of four MDR *P. aeruginosa* that is also resistant to the combination avibactam-ceftazidime (MICs = 40.4 µg/mL)). In all cases, the estimated MBC values doubled or overlapped those of the observed MICs. According to these values, the insignificant antibacterial effects of pristine CR232 (MICs ≥ 128 µg/mL) were improved by more than 3.2–25.3 times against bacteria of Gram-positive species and by more than 3.2–12.7 times against those of Gram-negative ones. Interestingly, the antibacterial effects of a series of 5-amido-1-(2,4-dinitrophenyl)-1H-4-pyrazolecarbonitriles, structurally like CR232 for the presence of a nitrile group on C4, were previously evaluated against bacterial collections. Notably, they were tested on American Type Cell Culture (ATCC) representatives of MRSA and MSSA, as well as against Persian Type Culture Collection (PTCC) representatives of *P. aeruginosa*, *B. subtilis*, and on an isolate of *E. coli* [30]. Curiously, the MICs observed on *P. aeruginosa* and *B. subtilis* were not reported. In any case, different from CR232 released by NPs, which proved to have remarkable antibacterial effects, all the reported compounds were inactive against *E. coli* (MIC > 400 vs. MICs = 20.2 µg/mL of CR232), while most compounds were less active than the CR232 released by CR232-G5K NPs against MRSA [30].

In another study conducted by Bekhit and Abdel-Aziem, a library of 12 pyrazole derivatives were again assayed on bacterial collections, including ATCC isolates of *E. coli* and *S. aureus,* observing MICs from 50 to > 200 µg/mL and MICs in the range 12.5- > 200 µg/mL, respectively. Considering only the most active pyrazole derivatives developed by Bekhit and colleagues (compounds **7** and **12a**), while **12a** displayed MIC = 50 µg/mL against *E. coli* and 25 µg/mL against *S. aureus*, **7** displayed MIC = 12.5 µg/mL against *S. aureus* and 100 µg/mL against *E. coli*. According to these results, even if less active than **7** against *S. aureus*, CR232 was 2.5-fold more active than **12a** and 5-fold more potent than **7**, on *E. coli* [31], thus establishing the high potency of the nanoengineered CR232 against bacteria of Gram-negative species, which, more than Gram-positive isolates, represent a global health concern [17]. 

More recently, a series of pyrazole derivatives containing 1,2,4-triazoles and benzoxazoles were shown to possess very potent antimicrobial activity against representative strains of *S. aureus* (MICs = 1.6–12.5 µg/mL)*, B. subtilis* (MICs = 1.6–25.0 µg/mL), *E. coli* (MICs = 3.1–25.0 µg/mL), and *P. aeruginosa* (MICs = 1.6–12.5 µg/mL). Nevertheless, the CR232 released from CR232-G5K NPs proved to be more active than several of these compounds. Particularly, on *B. subtilis,* CR232 was 1.2-, 2.5-, 2.5-, 1.2-, 2.5-, 1.2-, and 4.5-fold more potent than compounds identified as **12a**, **12b**, **12d**, **13a**, **13b**, **13c**, and **13d**, respectively. On *E. coli,* CR232 was 1.2-fold more potent than **12d**, while on *P. aeruginosa*, it was 1.2-fold more active than **12d**, **13b**, and **13d** [32]. 

Moreover, especially against *E. coli* and *P. aeruginosa*, CR232 released by the delivery system developed in this study displayed MBC values similar or lower than those obtained by the majority of a series of pyrazole–thiosemicarbazones (**3a**–**g**) and their pyrazole–thiazolidinone conjugates (**4a**–**g**) that were recently reported by Ebenezer and colleagues [33]. In particular, against *E. coli*, the MBCs estimated for CR232 were lower than those previously observed for compounds **3c**, **3d**, **3g**, and **4c** (MBCs = 20.2 µg/mL vs. MBCs = 137.9, 66.8, 69.6, and 150.5 µg/mL), while against *P. aeruginosa*, the MBCs of CR232 were lower than those observed for **3b**, **3e**, **3f**, **4a**, and **4c** (MBCs = 20.2 µg/mL vs. MBCs = 139.3, 71.5, 134.2, 162.6, and 37.6 µg/mL), where compounds **3b**, **3e**, **3f**, and **4a** were completely ineffective on *E. coli*. Additionally, CR232 showed MBC values about 4.5-fold lower than those obtained for compounds **3e** and **3g** against *K. pneumoniae*, and MBCs that were 1.7 and 1.9-fold lower than those determined for compounds **3b** and **4c** against MRSA. In any case, we make note that in the study by Ebenezer, as well as in all the other previous studies reviewed herein, the bacterial cells used to assay the pyrazole derivatives developed by the authors were always derived from ATCC or other bacterial collections. In this regard, we think that an overall merit of our study consists in having tested our compound on clinically relevant MDR bacterial isolates from infected patients. Interestingly, against *P. aeruginosa*, which represents one of the most frightening superbugs responsible of severe nosocomial infections associated to dramatic outcomes [17], the nanoengineered CR232 released by NPs displayed very low MIC values (MICs = 10.1, 10.1, 40.4 µg/mL, respectively) against one strain isolated from a patient affected by cystic fibrosis, one resistant to colistin, and one resistant to the recently developed association of avibactam-ceftazidime. Similarly, very low MICs (10.1 µg/mL) were also determined against a clinical isolate of KPCs-producing *K. pneumoniae* resistant to colistin, which is responsible for untreatable severe neonatal bacteremia.

The MIC values observed for CR232-G5K NPs and those estimated for the nanoengineered CR232 (free compound) delivered by the dendrimer formulation were compared to the MICs of commonly used antibiotics against the specific Gram-positive (Table 4) and Gram-negative (Table 5) pathogens. As the molecular weights (MW) of CR232-G5K NPs and antibiotics are very different, we compared the MIC values expressed as micromolar concentrations (µM), which provide how much of the equivalents of the substance under investigation have been administered to bacteria to obtain inhibition. Conversely, since CR232 (small molecule) released by NPs share similar MW values with the antibiotics, the comparison was carried out using the µg/mL scale, as per usual.

Accordingly, on all Gram-positive bacteria, the MICs (µM) of CR232-G5K NPs were exceptionally lower than those of the reference antibiotics, while those of CR232 released by NPs, expressed as µg/mL, were lower by 3.2–50.7 times.

Similarly, on all isolates of Gram-negative species, CR232-G5K NPs proved to be extraordinarily more potent than the reference antibiotics, while CR232 released from NPs, for all but two out of four isolates of KPCs-producing *K. pneumoniae,* displayed MICs lower by 1.6–6.3 times. Notably, both CR232-G5K NPs and the nanoengineered CR232 released by NPs emerged as active against colistin-resistant *P. aeruginosa* and *K. pneumoniae* strains with MIC values of 0.72 µM vs 18.5 µM of colistin (CR232-G5K NPs) and of 10.1 µg/mL vs. 16 µg/mL of colistin (CR232), respectively. Considering that colistin is the last therapeutic option against *P. aeruginosa* isolates resistant to all antibiotics these days, including carbapenem, and against carbapenem-resistant hypervirulent *K. pneumoniae* (CR-hvKP), and that, due to the emergence of colistin-resistant strains, will be soon no longer usable, the identification of a new antibacterial agent that is also active also against colistin-resistant *P. aeruginosa* and *K. pneumoniae* represents an exceptional achievement for this study.

#### 3.2.2. Time-Killing Curves

Time-kill experiments were performed with CR232-G5K NPs at concentrations equal to 4 x MIC on at least four strains for species of *P. aeruginosa*, *S. aureus*, and *E. coli*, including one colistin-resistant isolate of *P. aeruginosa* (strain 265) and one *P. aeruginosa* strain that is also resistant to the combination of avibactam-ceftazidime (strain 259), one isolate of *E. coli* producing NDMs (strain 462), and one MRSA (strain 187). As depicted in Figure 2, showing the curves obtained for the strains specified above, CR-232 G5K NPs displayed an extremely strong bactericidal effect against all the tested pathogens, causing an immediate and rapid decrease in the original cell number, which led to a total extinction of bacteria after only 2 h of exposure to CR232-G5K NPs. No significant difference among the various strains tested was observed, regardless of their specific resistances to different antibiotics. During the next two hours, and up to 24 h, no significant regrowth was observed for all the isolates, including the colistin-resistant *P. aeruginosa* isolate. Considering the difficulty in the treatment of *P. aeruginosa* strains that have developed resistance to colistin, the rapid bactericidal profile demonstrated by the CR232-formulation developed here must certainly be considered to have considerable interest for desirable clinical use.

It is noteworthy that, although the herbicidal and fungicidal effects of molecules containing the pyrazole ring have been reported [34,35,36], as to our knowledge, the only pyrazole-containing molecule known to possess bactericidal activity against certain Gram-negative and Gram-positive bacteria is ceftolozane, a semi-synthetic, broad-spectrum, fifth-generation cephalosporin antibiotic. Unfortunately, since inactivated by β-lactamase enzymes produced by several MDR bacteria, this drug is administered in combination with tazobactam, a β-lactamases inhibitor [27]. While tazobactam is capable of preventing ceftolozane inactivation by serine-type β-lactamases, it is unable to protect it from metallo-β-lactamases, thus establishing the inactivity of the ceftolozane/tazobactam combination against NDMs-producing bacteria [27]. Therefore, our work is the first study in which the bactericidal properties of a pyrazole-containing macromolecule has been successfully investigated by first measuring MBC values and then running time-kill experiments, which confirmed that CR232-G5K NPs possess a very potent and very rapid bactericidal effect on clinically relevant Gram-positive and Gram-negative strains, as well as MDR and even on isolates producing NDMs, against which, currently, no combination antibiotic/inhibitor is clinically approved [27].

### 3.3. Cytotoxicity of G5K, CR232, and CR232-G5K NPs on HaCaT Human Keratinocytes Cells

In addition to a proper water solubility, a new antibacterial agent should hopefully selectively inhibit the bacterial cell without damaging the eukaryotic one. This capability can be obtained by determining the values of selectivity index (SI) given by the ratio between the concentration of antibacterial agent capable to kill 50% of eukaryotic cells (LD_50_) and values of MICs. Thus, hoping for a possible cutaneous use of CR232-G5K NPs, a dose- and time-dependent cytotoxicity study was performed on human keratinocytes (HaCaT) to evaluate the effects on cell viability of pristine CR232, CR232-G5K NPs, and of the nano-manipulated CR232 provided by the quantity of NPs administered. HaCaT were selected as they are the principal type of cells found in the epidermis and are more susceptible to colonization by bacteria, fungi, and parasites. Since in our previous work, we have reported the cytotoxicity of the empty dendrimer G5K on HeLa cells [20], for comparison purposes, we herein evaluated the cytotoxicity of G5K on HaCaT cells. The cytotoxic activity of CR232, CR232-G5K NPs, and G5K at concentrations of 1, 5, 10, 15, 20, 25, 50, 75, and 100 µM was determined after 4, 12, and 24 h of exposure to the cells. The results are shown in Appendix A.

The cytotoxic effects of all compounds tested were strongly influenced by the exposure time. Thus, after 4 h, G5K was not cytotoxic even at the highest concentration tested (100 µM), leaving 88.5% of the cells alive and showing proliferation on the control (viability > 100%) at concentrations in the range 10–25µM. At this exposure time, the cytotoxicity of CR232 and that of CR232-G5K NPs was very similar. A slight proliferation was observed at 5–10 µM concentrations, while for higher concentrations, a slow decrease in cell viability was observed until reaching values of 41% viability at 100 µM concentration. After 12 h of exposure, the toxic effects were higher for all samples, with G5K being the most toxic compound and CR232 the least toxic one at concentrations in the range of 1–15 µM. At the highest concentration of 100 µM, the cell viability decreased under 50% for all compounds, evidencing very similar results for G5K and CR232 (40.4 and 40.8 %, respectively), while for CR232-G5K NPs, the cell viability was of 22.2%. After 24 h of exposure, the cytotoxicity of all compounds increased further, with CR232 being the least toxic compound at low concentrations (1–5 µM). At higher concentrations, the cytotoxicity of G5K and of CR232 was very similar, and for both compounds, the cell viability decreased to under 50% at a concentration of 25 µM to reach very low values at concentrations 100 µM. Curiously, for CR232-G5K NPs, the cell viability decreased to under 50% at a concentration of 5 µM, but at 100 µM, the cell viability evidenced a cytotoxicity lower than those of the empty dendrimer and of the pyrazole derivative.

With the aim of understanding whether, with its nanotechnological manipulation, the cytotoxicity of CR232 has been reduced or improved, and to calculate the SI values (LD_50_/MIC) of CR232, CR232-G5K NPs, and the nanoengineered CR232 provided by the dose of NPs administered, we reported the data of cell viability % obtained at 24 h of exposure vs. the concentrations of G5K, CR232, and CR232-G5K NPs. Next, based on the DL% value previously determined for CR232-G5K NPs (31.7%) [20], we estimated the concentrations of the nanoengineered CR232 provided by the formulation (41.2–4120.0 µM), which participated in the cytotoxic effects observed upon the administration of CR232-G5K NPs 1–100 µM. The obtained curves have been shown in Appendix A. Using appropriate parts of the curves in Appendix A and the equations of the regression models that best fit the related dispersion graphs, we determined the desired LD_50_. Figure 3a–c shows the dispersion graphs used, the best fitting regression models, and the related equations of G5K, CR232 (concentrations 1–100 µM), CR232-G5K NPs (concentrations 1–15 µM), and of the nanoengineered CR232 (concentrations 41.2–824.2 µM), used to compute their LD_50_.

The best fitting regressions models, which were polynomial for G5K and CR232, and exponential for CR232-G5K NPs and nanoengineered CR232 provided by NPs, were decided based on the value of the related coefficient of determination, R^2^. Moreover, since at concentrations of CR232-G5K NPs > 15 µM and of nano-manipulated CR232 > 824.2 µM the cell viability remained constant, data over these concentrations were not considered to obtain the related dispersion graph, their tendency lines, or their relative equations. The obtained equations, their R^2^ values, the computed values of LD_50_, and the range of SI obtained for the untreated CR232, for CR232-G5K NPs, and for the nanoengineered CR232 provided by NPs using Equation (1) have been reported in Table 6. The SI values of CR232-G5K NPs computed for each isolate used in this study have instead been reported and are observable in Table 2 and Table 3.
(1)$$\mathrm{SI}={\mathrm{LD}}_{50}/\mathrm{MIC}$$
were LD_50_ is the lethal dose (µg/mL or µM) of the antibacterial agent against HaCaT cells and MIC is the minimum inhibiting concentration (µg/mL or µM) displayed the same molecule against bacteria.

According to the LD_50_ data reported in Table 6, although the nano-formulation of CR232 developed here seems to be considerably cytotoxic, having a LD_50_ value of 5.6 µM (3.4-fold lower than that of G5K and 4.0-fold lower than that of untreated CR232), according to the MICs observed on isolates used in this study, its SI values (2–16) were remarkably higher than those determinable for pristine CR232, assuming MICs ≥ 128 µg/mL (SI ≤ 0.05789). In particular, SI values of CR232-G5K NPs were 34.5–276.4-fold higher than those of CR232, thus establishing that by formulating CR232 in NPs using G5K, a potent antibacterial agent was obtained that is more promising that the pristine pyrazole for being applied in therapy. Additionally, considering the LD_50_ values determined for the nanoengineered CR232 provided by the concentrations of NPs administered to the HaCaT cells (236.1 µM), it can be established that with our nanotechnological strategy, in addition to having improved the solubility of CR232 in water and its antibacterial effects, we have also reduced its cytotoxicity by 10.8 times. To date, the scientific community does not agree on the criterion for assessing the minimum acceptable value of SI. In fact, it was reported that SI values ≤ 5.2 were acceptable for South African plant leaf extracts with antibacterial properties, that antibacterial plant extracts were considered bioactive and non-toxic if SI > 1, and that SI should not be less than 2 [37,38,39,40]. Theoretically, the higher the SI ratio, the more effective and safer a compound would be during in vivo treatment for a certain bacterial infection. Collectively, the SI values determined in this study for CR232-G5K NPs (which are in the range 2–16) could be high enough to suggest they have a promising role as an antibacterial agent suitable for future clinical development.

## 4. Conclusions

A CR232-loaded dendrimer formulation (CR232-G5K NPs) was previously synthetized to solve the very poor solubility of CR232 that prevented the determination of reliable MICs and its possible safe administration in vivo. Here, the obtained NPs have been evaluated in vitro for their antibacterial effects on 36 strains, including MDR isolates of different species of Gram-positive and Gram-negative bacteria, with excellent results. Considerable antibacterial effects were observed against the most of the Gram-negative (MICs = 0.72–1.44 µM) and Gram-positive bacteria tested, with the best antibacterial effects being reported for *Enterococci* (MICs = 0.36–1.44 µM)*, S. epidermidis* (MICs = 0.36 µM), and *B. subtilis* (MIC = 0.36 µM).

Furthermore, CR232-G5K NPs also displayed very low MIC values (MIC = 0.72 µM) against colistin-resistant *P. aeruginosa* and *K. pneumoniae* isolates, which are currently untreatable by the available antibiotics. In addition, CR232-G5K NPs were effective against an isolate of *P. aeruginosa* resistant to the combination of avibactam-ceftazidime developed to counteract bacterial resistance to carbapenems.

Estimations of the MICs of the nanoengineered CR232 released by NPs at 24 h established MICs in the range 5.05–40.4 µg/mL against Gram-positive species, while MICs in the range 10.1–40.4 µg/mL were established against Gram-negative ones.

In all cases, both the MICs of CR232-G5K NPs and those estimated for the CR232 released by NPs were lower than those supposed for pristine CR232, which is difficult to determine due to its water insolubility, thus confirming that our nanotechnological approach not only succeeded in obtaining a water-soluble CR232 formulation suitable for both in vitro investigations and future in vivo applications, but also improved the antibacterial potency of CR232.

Additionally, in time-kill experiments carried out on strains of *P. aeruginosa*, *S. aureus*, and *E. coli*, including one colistin-resistant *P. aeruginosa,* one avibactam/ceftazidime-resistant *P. aeruginosa,* as well as NDMs-producing *E. coli* and MRSA strains, CR232-G5K NPs displayed an extremely strong and rapid bactericidal effect against all the tested pathogens, regardless their resistance to antibiotics. Finally, to evaluate the possible future clinical application of CR232-G5K NPs as a therapeutic agent, especially for skin infections, we examined its cytotoxicity on human keratinocyte cells (HaCaT) to determine the selectivity indices (SI) both of CR232-G5K NPs and CR232 provided by NPs upon administration of the dendrimer formulation. Accordingly, by formulating CR232 in NPs the SI values of pristine CR232 were improved by 34.5–276.4 times Additionally, according to the LD_50_ values determined for the nanoengineered CR232 provided by the concentrations of NPs administered to the HaCaT, by our nanotechnological strategy, the cytotoxicity of CR232 was reduced by 10.8 times. The results obtained in this study establish that, by formulating CR232 in NPs using G5K, a potent antibacterial agent more promising that the pristine pyrazole was obtained and could be applied in therapy.

## Figures and Tables

**Figure 1 biomedicines-10-00907-f001:**
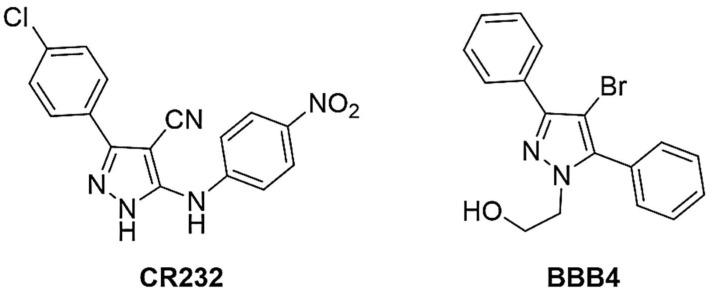
Chemical structure of CR232 and BBB4.

**Figure 2 biomedicines-10-00907-f002:**
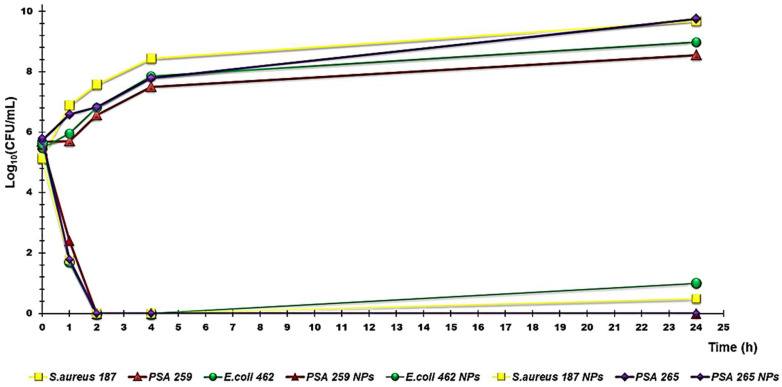
Time-killing curves performed with CR232-G5K NPs (at concentrations equal to 4 x MIC) on *P. aeruginosa* 265 and 259, *E. coli* 462, and *S. aureus* 187.

**Figure 3 biomedicines-10-00907-f003:**
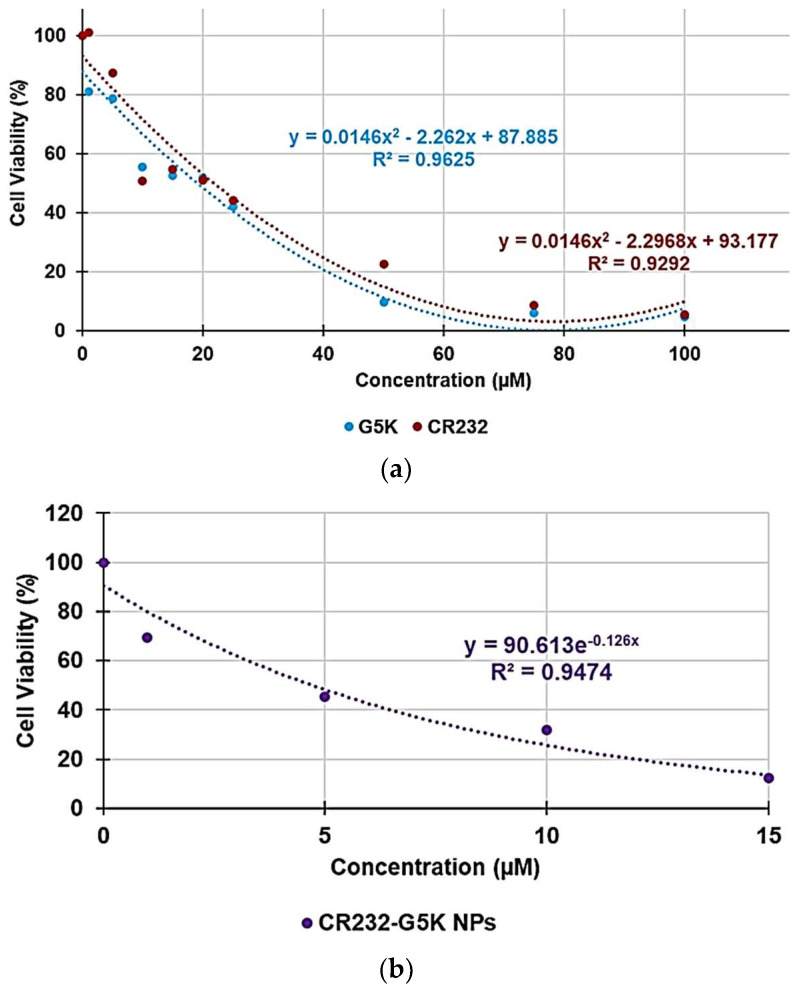
Regression models that better fit the dispersion graphs obtained when reporting the cell viability % vs. the concentration of samples at 24 h of exposure of CR232 and G5K in graph (**a**) of CR232-G5K NPs (**b**) and of the nano-formulated CR232 provided by the quantity of NPs administered (**c**).

**Table 1 biomedicines-10-00907-t001:** Main physicochemical properties of CR232-G5K NPs.

Analysis	CR232-G5K NPs	
FTIR [cm^−1^]	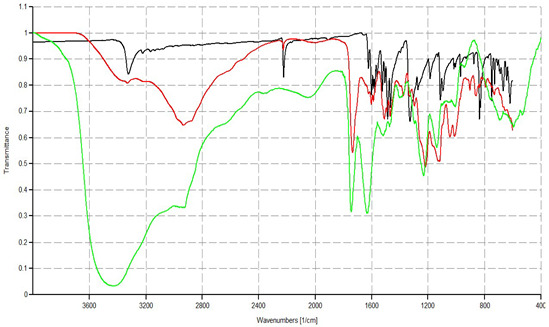 **G5K (green line), CR232 (black line), CR232-G5K NPs (red line)**	
^1^H NMR (400 MHz, DMSO-*d6*) [ppm]	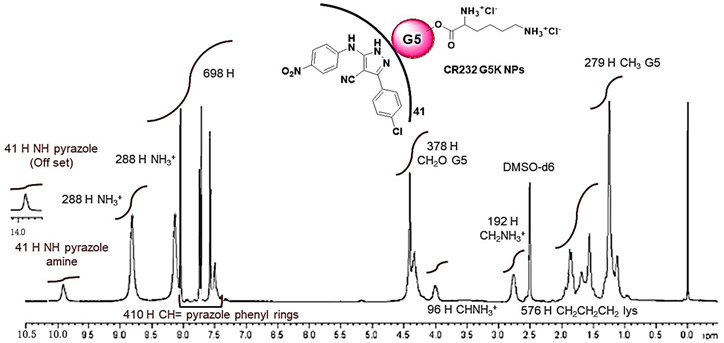	
^13^C NMR (100 MHz, DMSO-*d6*) [ppm]	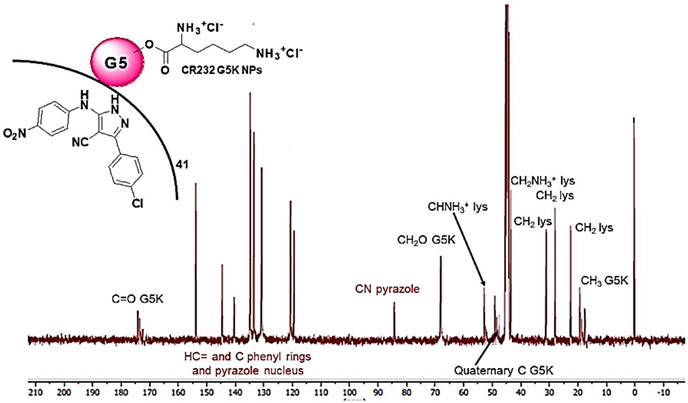	
UV-Vis	Ultraviolet Spectrum	ʎ _abs_ = 328 nm
UV-Vis	DL (%)	31.7 ± 0.6
	EE (%)	98.3 ± 2.0
^1^H NMR	MW	44153.1 44219.5 ± 237.8
DL% (UV-Vis)		
Scanning Electron Microscopy (SEM)	Morphology	Spherical
	Average Size	≃500 nm
DLS ^1^ Analysis	Z-Ave ^2^ (nm) PDI ^3^	529.7 ± 33.5 ^5^ 0.427 ± 0.054 ^5^
	ζ-*p* ^4^ (mV)	+37.2 ± 7.0 ^5^
Solubilization Essay	Water-Solubility (mg/mL)	5.2 ±0.05 ^6,§,8^ 1.65 ± 0.02 ^7^^,§,9^
Dialysis Method (UV-Vis)	Cumulative Release (%, 24 h)	99.3
	Mathematical Model	Weibull (β > 1)
	Mechanism	Complex Mechanisms
Cytotoxicity of G5K (HeLa Cells)	LD_50_	64.4 µM *
Potentiometric Titration	Buffer Capacity (β)	0.3076 0.1871
	Average Buffer Capacity (β _mean_)	

FTIR: the images show the spectra of empty delivery system G5K of CR232 and of CR232-G5K NPs; ^1^ dynamic light scattering; ^2^ hydrodynamic diameters of particles; ^3^ polydispersity indices; ^4^ measures of the electrical charge of particles suspended in the liquid of acquisition (water); ^5^ correspondent values for G5K = 175.7 ± 1.8, 0.129 ± 0.035, +48.0 ± 6.40; ^6^ refers to CR232-G5K NPs; ^7^ refers to CR232 contained in solubilized CR232-G5K; ^§^ water-solubility improvements: ^8^ 2311.1-fold; ^9^ 733.3-fold; * LD_50_ of paclitaxel = 22.4 µM and of G4-PAMAM-NH_2_ = 4.7 µM.

**Table 2 biomedicines-10-00907-t002:** MICs and MBCs of CR232-G5K NPs against bacteria of Gram-positive species, obtained from experiments carried out in triplicate ^1^, expressed as µM and µg/mL, and those of CR232 released according to the DL% and the release profile of CR232-G5K NPs [20], expressed as µg/mL.

	CR232-G5K NPs (44,220) ^2^		CR232 (339.7) ^2^		
Strains	MIC µM (µg/mL)	MBC µM (µg/mL)	MIC (µg/mL)	MBC (µg/mL)	Selectivity Indices ^3^
** *Enterococcus genus* **					
*E. faecalis* 1 *	0.72 (32)	1.44 (64)	10.1 20.2 10.1	20.2 20.2 20.2	8
*E. faecalis* 365 *	1.44 (64)	1.44 (64)			4
*E. faecalis* 439 *	0.72 (32)	1.44 (64)			8
*E. faecium* 21	0.36 (16)	0.72 (32)	5.05 5.05 5.05	10.1 10.1 10.1	16
*E. faecium* 300 *	0.36 (16)	0.72 (32)			16
*E. faecium* 369 *	0.36 (16)	0.72 (32)			16
** *Staphylococcus genus* **					
*S. aureus* 18 **	2.89 (128)	2.89 (128)	40.4	40.4	2
*S. aureus* 187 **	2.89 (128)	2.89 (128)	40.4	40.4	2
*S. aureus* 188 **	2.89 (128)	2.89 (128)	40.4	40.4	2
*S. aureus* 189 **	1.44 (64)	1.44 (64)	20.2	20.2	4
*S. epidermidis* 22 **	0.36 (16)	0.72 (32)	5.05	10.1	16
*S. epidermidis* 178	0.36 (16)	0.72 (32)	5.05	10.1	16
*S. epidermidis* 181 ***	0.36 (16)	0.72 (32)	5.05	10.1	16
** *Sporogenic isolate* **					
*B. subtilis*	0.36 (16)	0.36 (16)	5.05	5.05	16

^1^ the degree of concordance was 3/3 in all the experiments and the standard deviation (±SD) was zero; ^2^ MW; ^3^ refers to CR232-G5K NPs; * denotes vancomycin resistant (VRE); ** denotes methicillin resistant; *** denotes resistance toward methicillin and linezolid.

**Table 3 biomedicines-10-00907-t003:** MICs and MBCs of CR232-G5K NPs against bacteria of Gram-negative species, obtained from experiments carried out in triplicate ^1^, expressed as µM and µg/mL, and those of CR232 released according to the drug loading and the release profile of CR232-G5K NPs [20], expressed as µg/mL.

	CR232-G5K NPs (44,220) ^2^		CR232 (339.7) ^2^		
Strains	MIC µM (µg/mL)	MBC µM (µg/mL)	MIC (µg/mL)	MBC (µg/mL)	Selectivity Indices ^3^
** *Enterobacteriaceae family* **					
*E. coli* 224 S	0.72 (32)	0.72 (32)	10.1	10.1	8
*E. coli* 376#	1.44 (64)	1.44 (64)	20.2	20.2	4
*E. coli* 462 ^§^	0.72 (32)	0.72 (32)	10.1	10.1	8
*P. mirabilis* 254	>2.89 (128)	N.D.	>40.4	N.D.	<2
*M. morganii* 372	>2.89 (128)	N.D.	>40.4	N.D.	<2
*K. pneumoniae* 375#	1.44 (64)	2.89 (128)	20.2	40.4	4
*K. pneumoniae* 376#	2.89 (128)	2.89 (128)	40.4	40.4	2
*K. pneumoniae* 377#	2.89 (128)	2.89 (128)	40.4	40.4	2
*K. pneumoniae* 490#CR	0.72 (32)	2.89 (128)	20.2	40.4	8
*Salmonella gr. B* 227	2.89 (128)	2.89 (128)	40.4	40.4	2
*P. stuartii* 374	>2.89 (128)	N.D.	>40.4	N.D.	<2
** *Non-fermenting species* **					
*A. baumannii* 257	1.44 (64)	1.44 (64)	20.2	20.2	4
*A. baumannii* 279	0.72 (32)	0.72 (32)	10.1	10.1	8
*A. baumannii* 245	1.44 (64)	1.44 (64)	20.2	20.2	4
*P. aeruginosa* 1 V	0.72 (32)	1.44 (64)	10.1	20.2	8
*P. aeruginosa* 265 CR	0.72 (32)	1.44 (64)	10.1	20.2	8
*P. aeruginosa* 432	0.72 (32)	0.72 (32)	10.1	10.1	8
*P. aeruginosa* 259 *	2.89 (128)	2.89 (128)	40.4	40.4	2
*S. maltophilia* 466	0.72 (32)	1.44 (64)	10.1	20.2	8
*S. maltophilia* 390	0.72 (32)	1.44 (64)	10.1	20.2	8
*S. maltophilia* 392	1.44 (64)	2.89 (128)	20.2	40.4	4
*S. maltophilia* 392	0.72 (32)	0.72 (32)	10.1	10.1	8

^1^ the degree of concordance in all the experiments was 3/3 and the standard deviation (±SD) was zero; ^2^ MW; S denotes susceptibility to all antibiotics; ^3^ refers to CR232-G5K NPs; # denotes KPCs-producing *K. pneumoniae* isolates; § denotes New Delhi metallo carbapanemases (NDMs) producing isolate; *P. aeruginosa, S. maltophilia*, and *A. baumannii* were all MDR bacteria; 1 V = MDR strain isolated from a patient with cystic fibrosis; CR = MDR (*P. aeruginosa*) or KPCs-producing (*K. pneumoniae*) strains that are also resistant to colistin; * resistant to the combination avibactam-ceftazidime; N.D. = not detected.

**Table 4 biomedicines-10-00907-t004:** MIC values of CR232-G5K NPs and those estimated for the CR232 (free compound) delivered by the dendrimer formulation against bacteria of Gram-positive species, obtained from experiments carried out in triplicate ^1^, and those of reference antibiotics expressed as µM and µg/mL.

Strains	CR232-G5K NPs (44,220) ^2^	CR232 Released (339.7) ^2^	Reference Antibiotics
	MIC µM (µg/mL)	MIC µM (µg/mL)	MIC µM (µg/mL)
*E. faecalis* 1 *	0.72 (32)	29.7 (10.1)	193.2 (64) ^3^ 193.2 (64) ^3^ 193.2 (64) ^3^
*E. faecalis* 365 *	1.44 (64)	59.5 (20.2)	
*E. faecalis* 439 *	0.72 (32)	29.7 (10.1)	
*E. faecium* 21	0.36 (16)	14.9 (5.05)	700.6 (256) ^3^ 700.6 (256) ^3^ 700.6 (256) ^3^
*E. faecium* 300 *	0.36 (16)	14.9 (5.05)	
*E. faecium* 369 *	0.36 (16)	14.9 (5.05)	
*S. aureus* 18 **	2.89 (128)	119.0 (40.4)	386.4 (128) ^3^, 1401.2 (512) ^4^
*S. aureus* 187 **	2.89 (128)	119.0 (40.4)	386.4 (128) ^3^, 1401.2 (512) ^4^
*S. aureus* 188 **	2.89 (128)	119.0 (40.4)	386.4 (128) ^3^, 1401.2 (512) ^4^
*S. aureus* 189 **	1.44 (64)	59.5 (20.2)	386.4 (128) ^3^, 1401.2 (512) ^4^
*S. epidermidis* 22 **	0.36 (16)	14.9 (5.05)	193.2 (64) ^3^, 700.6 (256) ^4^
*S. epidermidis* 178	0.36 (16)	14.9 (5.05)	193.2 (64) ^3^, 700.6 (256) ^4^
*S. epidermidis* 181 ***	0.36 (16)	14.9 (5.05)	193.2 (64) ^3^, 700.6 (256) ^4^
*B. subtilis*	0.36 (16)	14.9 (5.05)	212.4 (128) ^5^

^1^ the degree of concordance was 3/3 in all the experiments and the standard deviation (±SD) was zero; ^2^ MW; * denotes vancomycin resistant (VRE); ** denotes methicillin resistant; *** denotes resistance toward methicillin and linezolid; ^3^ ciprofloxacin; ^4^ oxacillin; ^5^ amoxy-clavulanate.

**Table 5 biomedicines-10-00907-t005:** MIC values of CR232-G5K NPs and those estimated for the CR232 (free compound) delivered by the dendrimer formulation against bacteria of Gram-negative species (the isolated against which CR232-G5K NPs were inactive have been omitted), obtained from experiments carried out in triplicate ^1^, and those of reference antibiotics expressed as µM and µg/mL.

Strains	CR232-G5K NP (44,220) ^2^	CR232 Released (339.7) ^2^	Reference Antibiotics
	MIC µM (µg/mL)	MIC µM (µg/mL)	MIC µM (µg/mL)
*E. coli* 224 S	1.44 (64)	59.5 (20.2)	96.6 (32) ^3^
*E. coli* 376#	1.44 (64)	59.5 (20.2)	96.6 (32) ^3^
*E. coli* 462^§^	1.44 (64)	59.5 (20.2)	96.6 (32) ^3^
*K. pneumoniae* 375#	1.44 (64)	59.5 (20.2)	96.6 (32) ^3^
*K. pneumoniae* 376#	2.89 (128)	119.0 (40.4)	96.6 (32) ^3^
*K. pneumoniae* 377#	2.89 (128)	119.0 (40.4)	96.6 (32) ^3^
*K. pneumoniae* 490#CR	0.72 (32)	29.7 (10.1)	18.5 (16) ^4^
*Salmonella gr. B* 227	2.89 (128)	119.0 (40.4)	235.5 (128) ^5^
*A. bawmannii* 257	1.44 (64)	59.5 (20.2)	193.2 (64) ^3^
*A. bawmannii* 279	0.72 (32)	29.7 (10.1)	193.2 (64) ^3^
*A. bawmannii* 245	1.44 (64)	59.5 (20.2)	193.2 (64) ^3^
*P. aeruginosa* 1 V	0.72 (32)	29.7 (10.1)	76.2 (64) ^6^
*P. aeruginosa* 265 CR	0.72 (32)	29.7 (10.1)	18.5 (16) ^4^
*P. aeruginosa* 432	0.72 (32)	29.7 (10.1)	76.2 (64) ^6^
*P. aeruginosa* 259 *	2.89 (128)	119.0 (40.4)	76.2 (64) ^6^
*S. maltophilia* 466	0.72 (32)	29.7 (10.1)	117.7 (64) ^5^
*S. maltophilia* 390	0.72 (32)	29.7 (10.1)	117.7 (64) ^5^
*S. maltophilia* 392	1.44 (64)	59.5 (20.2)	117.7 (64) ^5^
*S. maltophilia* 392	0.72 (32)	29.7 (10.1)	117.7 (64) ^5^

^1^ the degree of concordance was 3/3 in all the experiments and the standard deviation (±SD) was zero; ^2^ MW; S denotes susceptibility to all antibiotics; # denotes carbapenemases (KPCs)-producing isolates; § denotes New Delhi metallo carbapanemases (NDMs) producing isolate; *P. aeruginosa, S. maltophilia* and *A. baumannii* were all MDR bacteria; 1 V = isolated from a patient with cystic fibrosis; CR = resistant to colistin; * resistant to combination avibactam-ceftazidime; N.R. = not reported; ^3^ ciprofloxacin; ^4^ colistin; ^5^ trimetoprim sulfametoxazole; ^6^ piperacillin tazobactam.

**Table 6 biomedicines-10-00907-t006:** Regression equations, R^2^ values, LD_50_ of G5K, CR232, CR232-G5K NPs, and of the nanoengineered CR232 provided by NPs according to the LD_50_ determined for CR232-G5K NPs (24 h treatments), as well as the relative SI ranges calculated using Equation (1).

Sample	Equations	R^2^	LD_50_ (µg/mL;µM)	SI
G5K	y = 0.0146x^2^ − 2.2620x + 87.8850	0.9625	577.40; 19.10	N.A.
CR232	y = 0.0146x^2^ − 2.2968x + 93.1770	0.9292	7.41; 21.83	≤0.05789
CR232-G5K NPs	y = 90.613e^−0.126x^	0.9474	247.6; 5.6	2–16 ^1^
CR232 provided by NPs	y = 92.160e^−0.003x^	0.9618	80.2; 236.1	2–16 ^1^

N.A. = not applicable; ^1^ the strains against which samples displayed MICs > 128 µg/mL were not considered.

## Data Availability

All data necessary to support reported results are present in the main text of the article and in the Appendix A.

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
