# Peer review of "Potent and Broad-Spectrum Bactericidal Activity of a Nanotechnologically Manipulated Novel Pyrazole"

_biomedicines, 2022, doi:10.3390/biomedicines10040907_

Round 1
Reviewer 1 Report
The manuscript titled "Strong and Broad-Spectrum Bactericidal Activity of a Novel Pyrazole Nanotechnologically Manipulated” (biomedicines-1673277) discusses the bactericidal effect of pyrazole-based nanoparticles against gram-positive and gram-negative bacteria. The submitted manuscript is quite an extent that may not be appealing to the reader. Herein, even though the introduction section is quite complete, I recommend the authors sum up the information provided. This section is too extensive. Nonetheless, the significance of the subject is well documented, as well as its novel character. It can be noted that the research team is well experienced in the present subject since it is clear that this work is preceded by previous studies.
Another concern that I have is referred to the number of auto citations which is around 35%. I understand that this work represents the validation of previously reported technology. Still, I considered this number too high and I strongly advise the authors to revise it.
The experiment design seems to be adequate and in line with what is commonly reported in related literature. The authors used a wide range of bacteria strains both gram-positive and negative. This section is concise and clear.
The results and discussion section begins with another extense introduction to the nanoparticles already synthesized and reported in a previous work which indeed facilitates its understanding. Section 3.2, however, could again be summarized. Given the size of the manuscript, it would be recommendable not to in some way repeat statements already presented in the introduction. This way, not only the manuscript would be clearer, but also, it would become more interesting for the reader.
The approach seems to present some novelty and considerable contribution to the state of the art, although ref 6, which is from the same authors, describes a similar study, using the same nanoparticles but against another bacteria. For this reason, the present manuscript should not be so insistent on establishing so many comparisons with other works. Seems that the authors keep stating the same things over and over again. This is especially repetitive since Ref 6 refers to a paper from 2022 published in Biomedicines, a Q1 journal. Gives the impression that is everything the same but with different bacteria. For this reason, I really think that the authors should rethink the present manuscript in order to turn it more appealing to the readers of Biomedicines' journal.
I advise the authors to rewrite the present manuscript and resubmit owing to the significance of the work. Considering the overall submission, I would not recommend the publication of this manuscript in its current form in the journal "Biomedicines".
Author Response
The manuscript titled "Strong and Broad-Spectrum Bactericidal Activity of a Novel Pyrazole Nanotechnologically Manipulated” (biomedicines-1673277) discusses the bactericidal effect of pyrazole-based nanoparticles against gram-positive and gram-negative bacteria. The submitted manuscript is quite an extent that may not be appealing to the reader. Herein, even though the introduction section is quite complete, I recommend the authors sum up the information provided. This section is too extensive. Nonetheless, the significance of the subject is well documented, as well as its novel character. It can be noted that the research team is well experienced in the present subject since it is clear that this work is preceded by previous studies.
We thank the Reviewer for having appreciated the significance of the subject of our study and our experience in the field, that sometimes leads us to be excessively verbose in our descriptions and in providing the reader with a complete background, thus risking being boring. So, in agreement with the Reviewer's suggestion, the Introduction section has been shortened by summing up the contained information. Please, check the revised Section 1 for such changes.
Another concern that I have is referred to the number of auto citations which is around 35%. I understand that this work represents the validation of previously reported technology. Still, I considered this number too high and I strongly advise the authors to revise it.
Again, we must admit that the Reviewer is right. Actually, without realizing it, due to the high number of studies we have carried out in the field of nanomedicine, we have exaggerated with self-citations. As suggested, self-citations have been reduced by half. In particular, the Ref. 2, 7, 9, 10, 11, 14 and 15 have been replaced with other references, relevant in the field
The experiment design seems to be adequate and in line with what is commonly reported in related literature. The authors used a wide range of bacteria strains both gram-positive and negative. This section is concise and clear.
We are very thankful to the Reviewer for having appreciate the experimental design of our work, and particularly the numerous microbiologic investigations we have carried out on several MDR bacteria of clinical relevance.
The results and discussion section begins with another extense introduction to the nanoparticles already synthesized and reported in a previous work which indeed facilitates its understanding.
We thank the Reviewer again for appreciating our effort to put the reader at ease by providing all the tools to be able to easily understand the derivation and the chemical-physical characteristics of the material herein biologically tested.
Section 3.2, however, could again be summarized. Given the size of the manuscript, it would be recommendable not to in some way repeat statements already presented in the introduction. This way, not only the manuscript would be clearer, but also, it would become more interesting for the reader.
Again, we must admit that the Reviewer is right. Accordingly, repetitions have been completely removed. Please, see the deletions at lines 277-280 and 290-300.
The approach seems to present some novelty and considerable contribution to the state of the art, although ref 6, which is from the same authors, describes a similar study, using the same nanoparticles but against another bacteria. For this reason, the present manuscript should not be so insistent on establishing so many comparisons with other works. Seems that the authors keep stating the same things over and over again. This is especially repetitive since Ref 6 refers to a paper from 2022 published in Biomedicines, a Q1 journal. Gives the impression that is everything the same but with different bacteria. For this reason, I really think that the authors should rethink the present manuscript in order to turn it more appealing to the readers of Biomedicines' journal.
We understand the Reviewer's reasoning and appreciate his desire to make the results reported in this study more stand out, by limiting the comparisons with those previously obtained with the antibacterial nanoparticles (NPs) recently published by our group just in Biomedicines. On the other hand, we are confident that such comparisons do not negatively affect the novelty of this study, nor make it seem less original in the eyes of Biomedicines readers. In fact, we kindly point out to the Reviewer, that, differently from what he has asserted in his comment, in our recent article published in Biomedicines (Alfei, et al. Pyrazole-Based Water-Soluble Dendrimer Nanoparticles as a Potential New Agent against Staphylococci. Biomedicines 2022, 10, 17. https://doi.org/10.3390/biomedicines10010017), we did not test the same nanoparticles reported in this new study, but totally different nanoparticles, made using a different dendrimer as encapsulating agent and a different pyrazole, which unlike CR232 already had a considerable per se antibacterial power. In that case, following its formulation in NPs, we increased both its solubility and its original antibacterial properties, which were however limited to Gram-positive MDR bacteria of Staphylococcus genus. In our opinion, to compare the antibacterial activity of CR232-G5K NPs of the present study with that of the pyrazole-based NPs recently reported by us, strongly highlights the microbiologic advances made in this new study. Notably, in this new study, using a new dendrimer as an encapsulating and solubilizing agent, we have "transformed" a practically inactive pyrazole, into a nanotechnologically manipulated compound with a potent antibacterial activity, since resulted effective on numerous strains of several species of both Gram-positive and Gram-negative bacteria, including colistin-resistant isolates of P. aeruginosa and K. pneumoniae, endowed also with rapid bactericidal activity not achieved with the previously tested particles. Additionally, we think that also comparisons with other works reporting on the antibacterial effects of previously studied pyrazole derivatives, are important to demonstrate the relevant achievements reached with this new study, particularly regarding the bactericidal activity, never reported for pyrazoles so far, except for ceftolozane.
On these considerations, we ask kindly the Reviewer to accept the style given to the discussion on the microbiologic results presented in this manuscript.
I advise the authors to rewrite the present manuscript and resubmit owing to the significance of the work. Considering the overall submission, I would not recommend the publication of this manuscript in its current form in the journal "Biomedicines".
As the Reviewer can observe the manuscript has been extensively revised and many parts completely rewrote to meet his requests. We hope that in this new form it could be considered suitable for publication in Biomedicines.
Reviewer 2 Report
Application of nanoparticle technology as a way to deal with drug hydrophobicity in the pyrazole family is promising research. I found the English writing to be awkward in places, but probably acceptable for an international journal. I have noted the following problems/questions concerning this manuscript:
Line 50: "New resistance mechanisms are emerging and spreading globally.." the references that follow promise to inform about new antimicrobial resistance mechanisms which would be interesting as most resistance is due to spread of already evolved mechanisms. Unfortunately the abstract of neither reference indicates anything about novel resistance mechanisms and instead simply describes other antibiotic advances made by these same authors. This seems inappropriate.
Line 52: "context", or "contest"? "Contest" would be unnecessarily anthropomorphic.
Line 146: In English, the word particle means the exact opposite of soluble. The physics of nanoparticles likely makes this distinction more problematic, but my first thought is that by definition, a nanoparticle is insoluble. Please examine your choice of words to be sure that the nanoparticles that have been constructed are truly "soluble" and not just hydrophilic, water-compatible, or some other term indicating long term, stable suspension in an aqueous environment.
Line 457-459: Reference 18 likely has several references to papers describing time-kill assays of pyrazole derivatives against Gram-positive bacteria. The statement in the text is incorrect.
Lines 468-469: The authors probably mean cell membranes. Mammalian cells do not have cell walls.
Line 496-502: In my experience with dose-response curves of cultured cells, the precision of low survival values at high drug concentrations is poor. Calculations based on viabilities between 0-15% seems inadvisable.
Lines 507-511: From what I understand looking at a previous reference, the modified SI values that have been calculated make use of the cumulative release values. However, these were previously calculated using a dialysis method. I would argue that the dynamics of this process could easily be altered by the presence of the protein-rich culture medium and particularly by the active cells in culture. I am highly sceptical of using the estimated release values to manipulate an SI calculation without evidence that the presence of living cells does not influence those values.
Author Response
Application of nanoparticle technology as a way to deal with drug hydrophobicity in the pyrazole family is promising research. I found the English writing to be awkward in places, but probably acceptable for an international journal.
On comment of this Reviewer, we have asked the help of the native English teacher Deirdre Kantz, working for both the University of Genoa and that of Pavia for revising the English language of our manuscript.
I have noted the following problems/questions concerning this manuscript:
Line 50: "New resistance mechanisms are emerging and spreading globally.." the references that follow promise to inform about new antimicrobial resistance mechanisms which would be interesting as most resistance is due to spread of already evolved mechanisms. Unfortunately the abstract of neither reference indicates anything about novel resistance mechanisms and instead simply describes other antibiotic advances made by these same authors. This seems inappropriate.
We apologise to the Reviewer for our inaccuracy. Ref. 10 and 11 have been replaced with more appropriate ones.
Line 52: "context", or "contest"? "Contest" would be unnecessarily anthropomorphic.
We apologise to the Reviewer for this typo. “Contest” has been replaced by “context”.
Line 146: In English, the word particle means the exact opposite of soluble. The physics of nanoparticles likely makes this distinction more problematic, but my first thought is that by definition, a nanoparticle is insoluble. Please examine your choice of words to be sure that the nanoparticles that have been constructed are truly "soluble" and not just hydrophilic, water-compatible, or some other term indicating long term, stable suspension in an aqueous environment.
We are confident that the used words are correct. The water-solubility of compounds both in the form of small molecules and of macromolecules depends mainly on their physicochemical characteristics, including dimensions, and it is known that smaller the size of the particles, the more their solubility increases. In this regard, several are the articles in literature reporting on the actual water solubility of appropriately structured nanoparticles. Please, consider some examples:
Sahoo, S., Dey, D. & Dhara, D. Water-soluble nanoparticles from PEGylated linear cationic block copolymers and anionic surfactants. Colloid Polym Sci 296, 183–193 (2018). https://doi.org/10.1007/s00396-017-4236-0.
Rakesh Banerjee, Saikat Maiti, Dibakar Dhara. Water-soluble nanoparticles from poly(ethylene glycol)-based cationic random copolymers and double-tail surfactant. Colloids and Surfaces A: Physicochemical and Engineering Aspects. Volume 395, 2012, Pages 255-261, ISSN 0927-7757,
https://doi.org/10.1016/j.colsurfa.2011.12.043.
Zinaida B. Shifrina†, Nina V. Kuchkina†, Pavel N. Rutkevich§, Tatyana N. Vlasik§, Anna D. Sushko‡, and Vladimir A. Izumrudov. Water-Soluble Cationic Aromatic Dendrimers and Their Complexation with DNA. Macromolecules 2009, 42, 24, 9548–9560.
In our specific case, the dendrimer nanoparticles (G5K) used to prepare CR232-G5K NPs possess a highly cationic surface due to the presence of 192 hydrochloride salt residues and then they are effectively highly soluble in water giving clear solutions even at high concentrations. Using such dendrimer NPs as encapsulating and solubilizing agent, we have prepared CR232-loaded NPs characterized by a surface presenting several amine groups in the form of hydrochloride salt as well, which confer to the complex pyrazole/dendrimer water solubility. Anyway, in line 146 (original manuscript) we only refer to a higher water solubility of dendrimer-based pyrazole-loaded NPs in comparison to liposomes-based ones.
Line 457-459: Reference 18 likely has several references to papers describing time-kill assays of pyrazole derivatives against Gram-positive bacteria. The statement in the text is incorrect.
We make kindly note to the Reviewer that if Ref. 18 would really have several references to papers describing time-killing assays of pyrazole derivatives against Gram-positive bacteria, such studies would be easily found with an accurate literature research, but they are not. In this regard, why does the Reviewer say “likely”? Is he not so sure? Anyway, we have extensively modified the text to correct our previous statement. Please, see lines 489-508.
Lines 468-469: The authors probably mean cell membranes. Mammalian cells do not have cell walls.
We apologise in advance to the Reviewer, but we do not understand his comment. Checking lines 468-469 we have found the following sentence:
“In addition to a proper water solubility, a new antibacterial agent should hopefully selectively inhibit the bacterial cell without damaging the eukaryotic one” which has nothing to do with what the Reviewer has commented. Additionally, using the function “search”, we have established that neither the word “wall” nor the word “mammalian” are present in our manuscript.
Line 496-502: In my experience with dose-response curves of cultured cells, the precision of low survival values at high drug concentrations is poor. Calculations based on viabilities between 0-15% seems inadvisable.
We thank the Reviewer for his comment which allowed us to avoid hazardous speculations. The assertions in lines 496-502 (original manuscript) have been modified accordingly. Please, see lines 538-544 (revised manuscript).
Lines 507-511: From what I understand looking at a previous reference, the modified SI values that have been calculated make use of the cumulative release values. However, these were previously calculated using a dialysis method. I would argue that the dynamics of this process could easily be altered by the presence of the protein-rich culture medium and particularly by the active cells in culture. I am highly sceptical of using the estimated release values to manipulate an SI calculation without evidence that the presence of living cells does not influence those values.
We thank the Reviewer for his useful comment. Accordingly, the micromolar concentrations of the nanotechnologically manipulated CR232 provided by the dose of CR232-G5K NPs which determined the observed values of cells viability, were estimated considering only the DL% of NPs. Consequently, Figure S9 (SM) and Figure 3c in the main text have been remade and some data in the main text have been modified. Then, considering the new equation associated at the regression model in Figure 3c, the LD50 of the treated CR232, was newly estimated. As the Reviewer can see, no significant differences were observed both for the value of LD50 and consequently for the SI range. Anyway, where necessary, the related discussion was modified accordingly.
Reviewer 3 Report
The manuscript entitled "Strong and Broad-Spectrum Bactericidal Activity of a Novel Pyrazole Nanotechnologically Manipulated" is a continuation of the previous work of the same authors. The results are important since they demonstrate the broad and wide spectrum of bactericidal activity of these CR232-G5K NPs even on some resistant strains. I would only have some suggestions for the authors:
- Maybe it would be better that you have some permission for the figures and tables related to the previously published data, otherwise, you risk being detected for plagiarism, because they are the same.
- It is quite hard to follow the paper since it brings information from previous work than from this work.
- Some sentences are too long and should be shortened, like the first sentence from the Conclusion section.
Author Response
The manuscript entitled "Strong and Broad-Spectrum Bactericidal Activity of a Novel Pyrazole Nanotechnologically Manipulated" is a continuation of the previous work of the same authors. The results are important since they demonstrate the broad and wide spectrum of bactericidal activity of these CR232-G5K NPs even on some resistant strains. I would only have some suggestions for the authors:
Maybe it would be better that you have some permission for the figures and tables related to the previously published data, otherwise, you risk being detected for plagiarism, because they are the same.
We thank the Reviewer for his suggestion, but we make kindly note that I am the first author of the previously published article containing data and images reported in Table 1 of the present study, and that all Figures and Table of this work are all my products. So, I need no permission. Please, see also Ref. 20.
It is quite hard to follow the paper since it brings information from previous work than from this work.
We make kindly note to the Reviewer, that to make easy the understanding of all information contained in the present work, including those reported in the previous correlated study, we have provided to the readers both a detailed Supporting Materials file and Table 1, summarizing important previously reported data on the physicochemical characteristics and on the synthesis of the CR232-G5K NPs, herein biologically evaluated.
Some sentences are too long and should be shortened, like the first sentence from the Conclusion section.
According to the Reviewer’s request too long sentences have been shortened including that signaled by the Reviewer (lines 608-613).
Round 2
Reviewer 1 Report
The authors have addressed all the questions I have made. Indeed, I strongly believe that the quality of the manuscript significantly increased. For this reason, I recommend the publication of the paper in its present form.
Author Response
The authors have addressed all the questions I have made. Indeed, I strongly believe that the quality of the manuscript significantly increased. For this reason, I recommend the publication of the paper in its present form.
We thank a lot the Reviewer for having appreciated the work of revision made by us, which was allowed mainly by his precious contribute, that has helped us in improving our manuscript ‘s quality.
Reviewer 2 Report
By careful rewording and editing, most of the problems in the manuscript have been circumvented. Despite improvement in language, there are still typos and awkward English construction in places, a matter between the editor and authors. This paper will be useful in moving the field forward.
Author Response
By careful rewording and editing, most of the problems in the manuscript have been circumvented. Despite improvement in language, there are still typos and awkward English construction in places, a matter between the editor and authors. This paper will be useful in moving the field forward.
We thank a lot the Reviewer for having appreciated the work of revision made by us, which was allowed mainly by his precious contribute, that has helped us in improving our manuscript ‘s quality.